# Early Knee Osteoarthritis Classification and Clinical Evolution: A Longitudinal Observational Pilot Study

**DOI:** 10.3390/biomedicines11061670

**Published:** 2023-06-09

**Authors:** Luz Herrero-Manley, Ana Alabajos-Cea, Luis Suso-Martí, Enrique Viosca-Herrero, Isabel Vazquez-Arce

**Affiliations:** 1Servicio de Medicina Física y Rehabilitación, Hospital La Fe, 46026 Valencia, Spain; 2Exercise Intervention for Health Research Group (EXINH-RG), Department of Physiotherapy, University of Valencia, 46010 Valencia, Spain

**Keywords:** osteoarthritis, knee osteoarthritis, early osteoarthritis, disability, diagnosis

## Abstract

Knee osteoarthritis (KOA) is one of the main problems of an aging society in terms of incidence, impairment to the quality of daily living (QOL), and economics. The main aim of this study was to verify the usefulness, in practical terms, of applying the existing diagnostic criteria of early knee osteoarthritis (EKOA). The secondary objective of this project was to evaluate the clinical progression of healthy subjects (HS) at risk of osteoarthritis and of patients with diagnosed EKOA. A cross-sectional longitudinal pilot study was carried out, in which 105 participants were classified as EKOA patients or HS according to the diagnostic criteria. Measures of disability, pain, and self-reported variables were assessed. Two follow-ups were performed in order to assess the diagnoses and radiological progression, and the clinical progression was evaluated using self-reported measures. Following the current diagnostic criteria, the participants were divided into EKOA and HS. Most of the participants did not present changes in their classification, although some subjects were reclassified as EKOA or HS in the follow-ups which were performed. The current classification criteria for EKOA based on self-reported measures, radiological findings, and clinical conditions such as pain could lead to a misdiagnosis of this process, as fluctuations in the classifications of patients according to their conditions were found during follow up.

## 1. Introduction

Osteoarthritis (OA) is one of the most significant causes of disability, and knee OA (KOA) in particular is one of the main problems for the elderly in terms of incidence, impairment to the quality of daily living (QOL), and economics [1]. KOA is a degenerative disorder which includes pathology of a wide range of tissues. It also involves structural and qualitative abnormalities that interfere with joint function, motor function, and physical activity [2]. KOA is an unavoidable pathological process for humans which is linked to the aging process; therefore, early diagnosis, treatment, and prevention should be the main focuses for the aging population [3].

During the last two decades, the term “early osteoarthritis” has emerged in the medical literature, with scientific papers on early osteoarthritis being increasingly published, and there is growing awareness of the importance of identifying the early stages of the degenerative processes in KOA [4]. It has been suggested that KOA progression may be prevented or delayed through early diagnosis before the joint is irreversibly damaged. When OA is treated appropriately at an early stage, therapeutic treatment or exercise may be effective in halting the progression of or even healing KOA [2].

Many descriptions of treatments for the early stages of KOA are available. For instance, chondroitin sulfate of non-animal origins is said to improve pain and knee function in EKOA [5].

Therefore, the importance of closely analyzing the latest diagnosis criteria cannot be emphasized enough. This is essential in order to verify their accuracy and to establish completely reliable criteria in the near future. With regard to prevention and treatment aspects, a clear definition of early knee OA (EKOA) is paramount for diagnosis [2].

In recent years, the definition of EKOA has begun to be discussed on a global scale, and various diagnostic criteria for identifying healthy subjects at risk of developing osteoarthritis have been put forward, in addition to criteria for detecting patients with EKOA [2,6]. The combination of the “symptoms” of joint pain; the “signs”, such as joint stiffness and tenderness; various risk factors; and methods of diagnostic imaging such as radiographs have all been proposed as ways to define early OA [2,6]. However, the definition and identification of EKOA is a complicated matter, as the signs/symptoms may still be limited and sporadic, only manifesting under certain conditions [7,8].

For example, in the early phases of OA, pain is related to activity, whereas in the later stages, pain becomes more constant over time [8]. In addition, it has been suggested that synovitis increases the risk of joint pain. Moreover, it is common to find signs of OA in the MRI results of asymptomatic patients. This finding highlights the fact that EOA changes can be asymptomatic, and pain and stiffness may only occur when the OA pathological process is already established [7,8].

Therefore, procuring a 100% definite diagnosis of EKOA is a well-known challenge. If a way could be found to make an accurate diagnosis of EKOA, the development of this disease could be slowed down considerably and, therefore, patients could be treated more promptly. This pilot study is innovative in that it applies the most recently accepted criteria for EKOA to a sample group of patients in order to verify their reliability in daily practice.

The main objective of this study was to verify the relevance and accuracy of EKOA diagnoses according to the existing criteria. Therefore, a cross-sectional longitudinal pilot study was carried out. After recruiting the patients, they were classified as healthy subjects (HS) or EOA using two diagnostic criteria, with the aim of improving early and reliable detection of patients with EKOA.

The criteria utilized were those of Luyten et al. and Mahmoudian et al. [9,10]. Both references include self-reported measures such as knee injury and osteoarthritis outcome (KOOS), radiological findings based on Kellgren and Lawrence (KL) classification, and clinical examination findings [9,10]. In addition, the secondary objective was to evaluate the clinical progression of healthy subjects at risk of OA, as well as patients with EKOA.

## 2. Materials and Methods

### 2.1. Study Design

A cross-sectional longitudinal pilot study with a non-probabilistic sample was carried out. Our patients were recruited and classified as healthy subjects (HS) or EOA, following the criteria of Luyten et al. and the classification of Mahmoudian et al. in order to attempt to improve the detection of EOA patients [9,10].

The study design adhered to the international recommendations for Strengthening the Reporting of Observational Studies in Epidemiology [11]. All participants were duly informed about the study procedures, which were planned according to the ethical standards of the Declaration of Helsinki and approved by the Ethics Committee on Drug Research of the hospital (CEIm 2017/0147). Each patient provided written informed consent, and the purpose of the study and the specific tests they would undergo were clearly explained.

### 2.2. Participants

The patients included in the study were selected at a hospital in Valencia (Spain), within the H2020 project OACTIVE. The design of the data collection protocol started on November 2017. Once set up, the recruitment of participants began in the summer of 2018, with follow-ups lasting until the end of February 2021. A total of 105 participants were divided into 2 groups, namely, EKOA and HS at risk of developing OA. The classification of the patients was based on two separate criteria so as to ensure a reliable diagnosis of EKOA.

First, Luyten’s criteria were applied [9], followed by those of Mahmoudian et al. [10]. All 105 participants were classified using both criteria, thus obtaining two separate samples per patient for follow-up (Figure 1 and Figure 2).

Luyten’s criteria were used to classify EKOA patients in the following ways:Patient-based questionnaires such as Knee Injury and Osteoarthritis Outcome score (KOOS) were used; 2 out of the 4 KOOS subscales (Symptoms, Knee pain, Function, and Knee-related quality of life) needed to score “positive” (≤85%);In the clinical examination, participants had to show crepitus or joint line tenderness;Regarding X-rays, patients with Kellgren and Lawrence (KL) grade 0–1 on standing weight-bearing radiographs (AP, lateral, AP fixed flexion, and skyline for patellofemoral OA) were included [9].

For the second analysis, patients were reclassified using the criteria laid out by Mahmoudian et al. [10].

In order to be classified as EKOA, subjects had to fulfill the following criteria:4.Patient-based questionnaires such as KOOS 4 score: the average of 4 of the 5 KOOS subscale averages, including pain, symptoms, ADL, and QOL (Pain, Symptoms, Function, and Knee-related quality of life) needed to score “positive” (≤80%);5.In the clinical examination, participants had to show crepitus or joint line tenderness.6.Regarding X-rays, patients with KL grade 0–1 on standing weight-bearing radiographs (AP, lateral, AP fixed flexion, and skyline for patellofemoral OA) were included.

There were three inclusion criteria for HS at risk of OA:-Kellgren and Lawrence grade 0–1.-Regarding age limit, patients were required to be 40 years old or over.-Participants had to be overweight, with a body mass index of at least 25.

The employed exclusion criteria were identical for EKOA and HS at risk of developing OA, and comprised autoimmune, rheumatic, and infectious conditions. Additionally, patients suffering from cognitive impairments which hindered their viewing of the audio–visual material and those with illiteracy, comprehension or communication issues, or insufficient knowledge of Spanish for following measurement instructions were also excluded.

### 2.3. Outcome Measures

#### 2.3.1. Demographic Data and Control Variables

The study incorporated general demographic information obtained from extensive databases, along with well-established conventional predictors such as gender, age, educational level, and marital status [12,13]. Additionally, data regarding smoking and alcohol consumption habits were registered. In the case of women, hormonal status was also taken into account. Finally, measurements of weight, height, and QOL were also collected.

#### 2.3.2. Pain and Disability Variables

##### Pain Intensity

In order to quantify pain intensity, the Visual Analogue Scale (VAS) was used. The VAS is a horizontal line 100 mm in length with verbal descriptors (word anchors) at each end to express the extremes of painful sensations (ranging from no pain to the maximum pain imaginable). Patients mark the point on the line that best corresponds to the severity of their symptoms. It has been shown to have good retest reliability (r = 0.94, *p* < 0.001) and a minimal detectable change of 15.0 mm [14,15]. The VAS scale was applied both at rest and while walking.

##### Western Ontario and McMaster Universities Osteoarthritis Index (WOMAC)

This questionnaire is widely used to evaluate the conditions of patients with osteoarthritis; it assesses the functional and symptomatic features.

The WOMAC questionnaire is self-administered and is used to assess patients with progressive hip and/or knee osteoarthritis. The questionnaire uses a multidimensional scale composed of 24 items divided into 3 aspects: functional pain (consisting of 5 items), stiffness (2 items), and difficulties in daily activities (17 items). Higher values indicate poorer scores for pain and physical function on the WOMAC subscales. The Spanish version of the WOMAC questionnaire has adequate psychometric properties, presenting an index of internal consistency (a) of 0.82 for pain and 0.93 for physical function [16].

##### Knee Injury and Osteoarthritis Outcome Score (KOOS)

The KOOS is a knee-specific questionnaire designed to assess short- and long-term patient-relevant opinions regarding their knee issues and other associated problems. The KOOS is self-administered and assesses five outcomes: Pain, other Symptoms, Function in daily living (ADL), Function in Sport and Recreation (Sport/Rec), and Knee-related Quality of Life (QOL). The psychometric properties and the ICC of the Spanish version have shown acceptable properties for both the total score and the subscales [17].

#### 2.3.3. Psychological Variables

##### Anxiety and Depression Symptoms

The Spanish version of the Hospital Anxiety and Depression Scale was applied to identify depression and anxiety symptoms in the participants [18]. This scale includes 14 items, which are rated on a 4-point Likert-type scale. Two subscales assessed depression and anxiety independently. The internal consistency was 0.90 for the full scale; 0.84 for the depression subscale; and 0.85 for the anxiety subscale [18].

### 2.4. Procedures

Every participant was provided with an information sheet elucidating the procedure, together with a consent form to ensure that they were well-informed. Subsequently, participants were urged to express any inquiries pertaining to the nature of the study.

The subjects who agreed to participate then proceeded to fill in the sociodemographic questionnaire. Self-reported measures of disability, pain, and any variables were then assessed. The initial follow-up took place between July 2020 and September 2020, while the second follow-up occurred between January 2021 and February 2021.

### 2.5. Statistical Analysis

The statistical data analysis was conducted using statistical SPSS software, version 22.0 (SPSS Inc., Chicago, IL, USA). The normality of the variables was evaluated using the Shapiro–Wilk test. Descriptive statistics were used to summarize the data for continuous variables, and are presented as mean ± standard deviation, 95% confidence interval. A 2-way repeated measures analysis of variance (ANOVA) was conducted to study the effects of the between-participant “group” factor in each of the two categories (EOA and HS) and the within-participant “time” factor in each of the three categories (i.e., pre-, post-1, and post-2).

A post hoc analysis with Bonferroni correction was carried out in the case of significant ANOVA findings for multiple comparisons between variables. Effect sizes (d) were calculated in accordance with Cohen’s method, in which the magnitude of the effect was classified as small (0.20–0.49), moderate (0.50–0.79), or large (0.8) [19].

## 3. Results

Altogether, 105 subjects were included (54 EKOA and 51 HS). A total of 11 HS group patients had been lost by the first follow-up, and 14 by the second round. As for the EKOA group, 15 subjects had dropped out by the first follow-up and 11 by the second (Figure 1). The main reason for this was the COVID-19 pandemic, but several patients did not wish to continue for personal reasons.

All the variables presented normal distribution. No statistically significant differences were found between groups for any of the primary variables, demographic data, or self-report variables at baseline (Table 1).

### 3.1. Evolution Classification

Following the Luyten et al. [9] diagnostic criteria described above, participants were reclassified at each follow-up, which were between 9 to 12 months apart. In the EOA group, 4 participants at the first follow-up and another 4 participants at the second follow-up presented an improvement that caused their classification to change to HS. Conversely, 18 HS participants evolved to EOA at the first follow-up, but 6 of them were again classified as HS at the second follow-up (Figure 1). None of the participants progressed to advanced OA.

We also reclassified the patients during each follow-up according to the suggestions of Mahmoudian et al. [10]. Initially, the 105 subjects were classified as 44 EOA and 61 HS. In the EOA group, 5 subjects at the first follow-up and 2 at the second follow-up presented an improvement that changed their classification to HS. In addition, 14 HS evolved to EOA at the first follow-up (1 of them being classified as HS again at the second follow-up), and 2 at the second follow-up (Figure 2). None of the participants progressed to advanced OA.

### 3.2. Clinical Progression

#### 3.2.1. Pain Intensity

The ANOVA did not reveal any statistically significant changes in VAS rest measurement overtime (*F* = 1.89, *p* = 0.16, ƞ^2^ = 0.79). It showed no differences between baseline, follow-up 1, and follow-up 2 for either the EOA or the HS group. Similar results were found using the classification of Mahmoudian et al. [10] (Table 2 and Table 3).

The ANOVA did not reveal any statistically significant changes in the VAS walking measurement overtime (*F* = 0.051, *p* = 0.61, ƞ^2^ = 0.02). It showed no differences between baseline, follow-up 1 and follow-up 2 for EOA or HS groups. Similar results were found using the classification of Mahmoudian et al. [10] (Table 2 and Table 3).

#### 3.2.2. Disability

The ANOVA revealed statistically significant changes in WOMAC measurement overtime (*F* = 3.10, *p* = 0.048, ƞ^2^ = 0.181), showing differences between the baseline and follow-up 2 for the EOA and HS groups. Similar results were found using the classification of Mahmoudian et al. [10] (Table 2 and Table 3).

#### 3.2.3. Psychological Variables

The ANOVA did not reveal any statistically significant changes in HAD Anxiety measurement overtime (*F* = 0.14, *p* = 0.86, ƞ^2^ = 0.01), nor in HAD Depression (*F* = 0.66, *p* = 0.52, ƞ^2^ = 0.02); no differences were found between baseline, follow-up 1, and follow-up 2 for the OA or the HS group.

Finally, the ANOVA did not reveal statistically significant changes in any of the KOOS subscale (pain, symptoms, QQL, ADL, and sport subscales) measurements overtime for the EOA or the HS group. Similar results were found using the classification of Mahmoudian et al. [10] (Table 2 and Table 3).

## 4. Discussion

### 4.1. Diagnosis Criteria of EOA

With the objective of evaluating the reliability of EKOA diagnosis and the clinical progression of our patients, we used Luyten’s criteria to classify them as HS or EOA. These classification criteria were intended only for research, as the authors declare, and showed a specificity of 76.5% for the detection of clinical progression [9,10].

At present, very few studies have evaluated the progression of patients with EKOA. In this regard, we found that some HS met the criteria for EKOA diagnosis at a certain moment in time who, afterwards, scored as HS again; we also found subjects classified as EKOA at the beginning who scored as HS at one or both follow-ups. It should be noted that these criteria are mainly based on clinical condition, so this finding made us wonder if it was possible that we were not actually detecting changes related to the onset of OA, but knee pain due to other reasons, leading to misdiagnoses. These findings may suggest that in order to diagnose EKOA in a more accurate way, new diagnostic criteria could be considered, such as biomechanical parameters, WOMAC criteria, or pain persistence, during a longer established period of time. This is to be the subject of a future study.

Recently, Mahmoudian et al., 2020, evaluated the classification criteria for EKOA, attempting to find some aspects that could refine the diagnosis [10]. Their results showed how these criteria can contribute to the prediction of structural and clinical progression (sensitivity of 29% and 43.5%, specificity of 73.1% and 76.5%, respectively). In addition, these authors noted that for the purpose of detecting structural evolution, a K&L grade I could be a better predictor. In relation to clinical evolution, it seems that adding more clinical aspects, such as effusion or Heberden’s nodes, could increase the predictive value of the criteria, emphasizing the importance of physical examination. They found the best predictive performance for KOOS4 ≤ 80% [10]. We analyzed our sample again considering these suggestions, mainly the KOOS4 ≤ 80%, as we had a very small sample of subjects with K&L I, and none of our subjects had Heberden’s nodes. After applying these modifications, we still found some changes from the EKOA to the HS group at the follow-ups, but they were fewer than the initial numbers.

These fluctuations between HS and EKOA patient classification in the follow-up using both criteria could point to a certain degree of unreliability in the diagnosis of EKOA, which raises doubts concerning the use of this classification in practice and highlights the clinical relevance of this pilot study.

### 4.2. Clinical Progression of OA

Our data are in keeping with previous results in the field of musculoskeletal pain and, specifically, in OA [20,21]. Regarding this, patients with OA were found to show fluctuations and exacerbations directly related to a linear progression of the disease or to the radiographic development of OA [22,23].

It is possible that EKOA may not be progressive for some time, and also that the clinical course depends on individual epigenetic, neurophysiological, or metabolic factors which need to be studied further [24,25]. In addition, there was no impairment of functional capacity due to pain or anticipation of pain, something that has been suggested previously [26,27]. In fact, our results also showed fluctuations in the functional statuses of OA patients, in agreement with some previous studies [28]. Only the WOMAC measurement showed fluctuations over time, showing differences between the baseline and follow-up for the EOA and HS groups.

Future studies with a longer period of follow-up, and even studies that include new factors such as, for example, biomechanical parameters, should address this question and determine whether there are indeed clinical variables that could determine EKOA. Nevertheless, in the interest of practicality, the follow-up period should not be overly extended.

## 5. Limitations

As this is a pilot study, there are certain limitations to be taken into consideration. Furthermore, due to the cross-sectional design of this study, it is not feasible to establish a causal relationship. In addition, a more extended period of follow-up is likely required in order to assess the progression of these patients and to validate our findings. In addition, as discussed above, we used the existing classification criteria to divide our patients into HS and EOA groups in order to evaluate their clinical progression. However, because of the variability in this classification overtime, we cannot be sure that this classification is correct.

## 6. Conclusions

The findings of the current study suggest that the existing classification criteria for EKOA based on self-reported measures, radiological findings, and clinical conditions such as pain could lead to misdiagnosis, as they raise doubts concerning practical viability. Furthermore, our results show that HS at risk of OA and patients diagnosed as EKOA according to the current classification criteria, did not show any significant changes in pain data, disability, or psychological variables for this follow-up period.

## Figures and Tables

**Figure 1 biomedicines-11-01670-f001:**
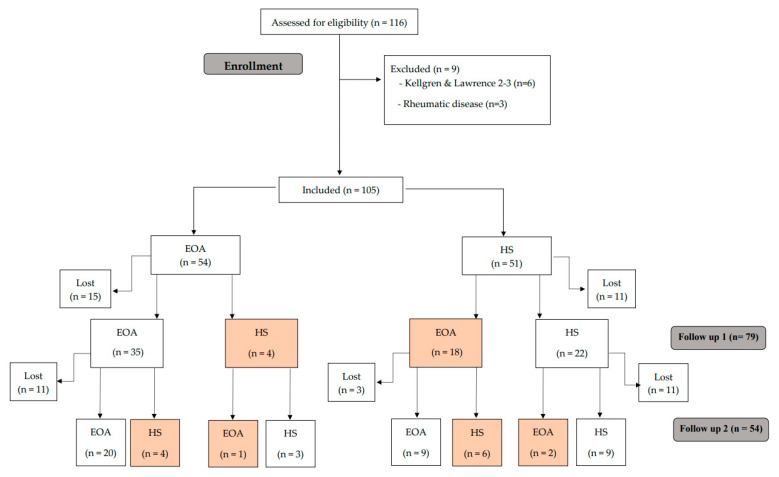
Study flow chart following the criteria of Luyten et al. [9].

**Figure 2 biomedicines-11-01670-f002:**
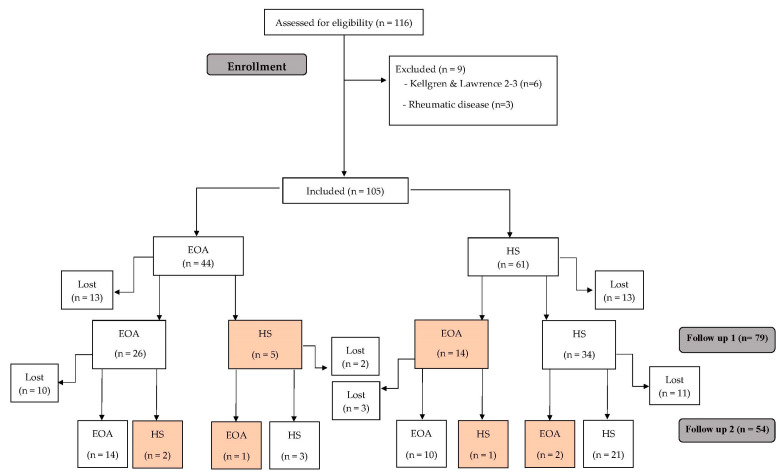
Study flow chart following the criteria of Mahmoudian et al. [10].

**Table 1 biomedicines-11-01670-t001:** Descriptive and demographic data, as well as control variables.

	EOA (*n* = 54)	HS (*n* = 51)	*p* Value
Age (years)	51.85 ± 5.72	50.49 ± 6.21	0.46
BMI (kg/m^2^)	27.40 ± 4.19	26.80 ± 3.68	0.43
Gender			0.17
Women	35 (64.8)	35 (68.6)	
Men	19 (35.2)	16 (31.4)	
Marital status			0.95
Single	5	7	
Married	31	30	
Widowed	2	2	
Divorced	6	6	
Level of formal education			0.18
Primary	6	9	
Secondary	20	12	
College	18	24	
Smoking			0.27
No	18	23	
Yes	9	4	
Ex	21	19	
Alcohol			0.65
Never	8	4	
Seldom	13	18	
1–2 times/month	11	9	
1–2 times/week	15	15	
1 time per day	2	2	
More than 1 per day	1	0	

**Table 2 biomedicines-11-01670-t002:** Clinical progression of OA patients.

	Baseline	Follow-Up 1	Follow-Up 2
**VAS Rest**			
Luyten et al. [9]	2.04 ± 1.56	1.29 ± 2.05	2.21 ± 2.67
Mahmoudian et al. [10]	1.88 ± 2.31	1.50 ± 2.06	2.44 ± 2.78
**VAS Walking**			
Luyten et al. [9]	2.67 ± 2.73	2.42 ± 2.02	2.46 ± 2.40
Mahmoudian et al. [10]	2.69 ± 2.77	2.94 ± 2.11	2.63 ± 2.55
*** WOMAC**			
Luyten et al. [9]	18.55 ± 10.15	16.56 ± 15.66 *	22.47 ± 17.23 *
Mahmoudian et al. [10]	17.62 ± 9.47	22.04 ± 16.75 *	25.26 ± 18.75 *
**KOOS Pain**			
Luyten et al. [9]	80.48 ± 13.44	78.62 ± 16.24	80.52 ± 19.97
Mahmoudian et al. [10]	86.17 ± 14.31	75.08 ± 19.40	74.25 ± 23.42
**KOOS Symptoms**			
Luyten et al. [9]	85.00 ± 12.71	79.62 ± 15.81	83.61 ± 12.45
Mahmoudian et al. [10]	85.92 ± 12.41	74.24 ± 16.58	77.67 ± 12.07
**KOOS ADL**			
Luyten et al. [9]	83.29 ± 16.94	79.62 ± 18.53	81.95 ± 18.67
Mahmoudian et al. [10]	86.83 ± 18.66	73.67 ± 21.62	73.92 ± 20.91
**KOOS QQL**			
Luyten et al. [9]	62.29 ± 26.56	51.86 ± 20.45	61.33 ± 24.59
Mahmoudian et al. [10]	65.17 ± 32.80	45.42 ± 18.64	50.08 ± 22.37
**KOOS Sport**			
Luyten et al. [9]	60.71 ± 30.83	55.71 ± 29.43	57.61 ± 26.77
Mahmoudian et al. [10]	58.75 ± 37.54	44.58 ± 30.63	46.25 ± 24.23
**HAD Anxiety**			
Luyten et al. [9]	5.42 ± 4.14	4.17 ± 3.81	5.29 ± 4.60
Mahmoudian et al. [10]	4.94 ± 3.35	4.75 ± 2.46	4.88 ± 3.20
**HAD Depression**			
Luyten et al. [9]	3.38 ± 3.07	3.13 ± 2.91	2.96 ± 3.32
Mahmoudian et al. [10]	2.81 ± 2.34	3.31 ± 2.85	2.50 ± 2.68

* Statistical significance. Data are presented as mean ± SD.

**Table 3 biomedicines-11-01670-t003:** Clinical progression of healthy subjects at risk of developing OA.

	Baseline	Follow-Up 1	Follow-Up 2
**VAS Rest**			
Luyten et al. [9]	0.1 ± 0.29	0.74 ± 1.36	0.91 ± 1.41
Mahmoudian et al. [10]	0.68 ± 1.85	0.77 ± 1.54	1.13 ± 1.77
**VAS Walking**			
Luyten et al. [9]	0.52 ± 1.76	1.3 ± 2.01	1.52 ± 1.86
Mahmoudian et al. [10]	1.06 ± 2.23	1.32 ± 1.85	1.68 ± 1.92
*** WOMAC**			
Luyten et al. [9]	5.86 ± 4.3	9.57 ± 8.20 *	12.30 ± 10.30 *
Mahmoudian et al. [10]	9.32 ± 8.89	12.44 ± 8.45 *	12.77 ± 10.42 *
**KOOS Pain**			
Luyten et al. [9]	88.80 ± 13.05	87.60 ± 9.65	88.96 ± 12.28
Mahmoudian et al. [10]	84.10 ± 14.31	86.90 ± 9.31	87.47 ± 11.61
**KOOS Symptoms**			
Luyten et al. [9]	90.24 ± 10.83	91.80 ± 6.79	92.60 ± 8.80
Mahmoudian et al. [10]	88.20 ± 12.20	90.53 ± 7.84	91.30 ± 8.51
**KOOS ADL**			
Luyten et al. [9]	89.88 ± 14.37	90.32 ± 7.84	92.01 ± 9.78
Mahmoudian et al. [10]	86.13 ± 15.54	89.83 ± 7.66	91.20 ± 9.06
**KOOS QQL**			
Luyten et al. [9]	76.88 ± 22.29	75.12 ± 20.62	79.64 ± 21.88
Mahmoudian et al. [10]	71.57 ± 22.72	71.97 ± 20.53	75.93 ± 20.81
**KOOS Sport**			
Luyten et al. [9]	71.40 ± 26.71	71.60 ± 22.16	72.60 ± 23.29
Mahmoudian et al. [10]	67.67 ± 25.72	71.33 ± 21.01	69.83 ± 22.03
**HAD Anxiety**			
Luyten et al. [9]	2.83 ± 3.34	3.04 ± 3.34	2.87 ± 2.79
Mahmoudian et al. [10]	3.74 ± 4.21	3.45 ± 4.09	3.71 ± 4.31
**HAD Depression**			
Luyten et al. [9]	1.17 ± 2.05	1.52 ± 2.27	1.26 ± 1.18
Mahmoudian et al. [10]	2.03 ± 3.04	1.84 ± 2.54	1.94 ± 2.62

* Statistical significance. Data presented as Mean ± SD.

## Data Availability

Not applicable.

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
