# Peer review of "Early Knee Osteoarthritis Classification and Clinical Evolution: A Longitudinal Observational Pilot Study"

_biomedicines, 2023, doi:10.3390/biomedicines11061670_

Round 1

Reviewer 1 Report

The manuscript describe and exploit the results of a  cross-sequential (cross-sectional longitudinal) study dedicated to the clinical progression of knee osteoarthritis, conducted on 105 human subjects (of which about half were healthy but at risk; some of them were lost from their group, for objective reasons, during the study), aiming at earlier and better diagnosis based on relevant and accurate criteria. Two diagnostic criteria were applied (i) for inter-groups comparisons, (ii) to evaluate the evolution of the disease, as well as for (iii) reclassification reasons. Accordingly, two rationally organized (even if rather complex) study flow-charts were conceived. The Limitations section is well written and appropriately evasive.

General remarks on the manuscript content

G1. The manuscript seems to fit poorly into the area of interest of the Biomedicines journal. To increase the interest of the journal, please follow remarks G2 and G3. Please consider that you are not addressing only the statisticians or epidemiologists.

G2. A better way of describing and commenting the philosophy of the dynamically designed cross-sequential study should inserted in the introductory section of the manuscript, in order to introduce the readers of Biomedicines journal into the complex landscape of modern biostatistics.

G3. At least, a comparative comment should be inserted to discriminate between the two diagnostic / (re)classification criteria (Luyten and Mahmoudian) which were used in the study. 

Pointwise remarks on the manuscript content 

R1. Reference 18 should be correctly mentioned in the References section.

R2. Some of the concrete descriptions in Section 3.1 should be "prepared" by offering their (bio)statistical and epidemiological basis, into the introductory section. As an example, the rationale of reclassifications you have performed.

R3. The Discussion section abounds in clinically relevant information, but is poorly related with section 3.

English used to write the manuscript is acceptable. Some corrections of punctuation and phrasing are still required.

Author Response

Dear Sir/Madam,

Thank you for your suggestions for improving the paper.

We have taken them into consideration and made changes according to your suggestions.

In the introduction section a new paragraph describing  the design of the study has been added.  We have also included a comment on the criteria used in the introduction. This is in addition to the fact  that they are described in more detail in the Materials and Methods section.

We have corrected reference 18 and it is now number 19, as more biography references have been added.

In addition, the  standard of English has been checked and improved.

Please do not hesitate to inform me of any further changes which may be required.

Thank you for all your help.

Best regards,

Luz Herrero Manley

Reviewer 2 Report

This is a very interesting paper, well written and with a medium scientific level.

In overall this paper deserves a good evaluation. 

I have minor comments:

1) why in table 1 is not reported the p value for smoke and alchool consumption?

2) please describe the possible alternative therapies for K.O reccommended in this paper and cite it.

Rondanelli, M., Braschi, V., Gasparri, C., Nichetti, M., Faliva, M.A., Peroni, G., Naso, M., Iannello, G., Spadaccini, D., Miraglia, N. and Putignano, P., 2019. Effectiveness of non-animal chondroitin sulfate supplementation in the treatment of moderate knee osteoarthritis in a group of overweight subjects: a randomized, double-blind, placebo-controlled pilot study. Nutrients11(9), p.2027.

4) table 3. Unit of measures are missing

5) p value over time is missing

6) Data have been reported as mean and sd but they did not show this in table

7) figures are missing. Please diplay in figure the main results statisticially significnt

quality could be improved

Author Response

Dear Sir/Madam,

Thank you for your suggestions for improving the paper.

We have taken them into consideration and made changes according to your suggestions.

We have added the  p value for smoke and alcohol consumption to table 1.

In addition,  the possible alternative therapies for knee OA described in  the paper you recommended have been discussed.

We have  also added the missing values and other items which were lacking.

Regarding the  missing figures you mentioned, as the paper already has two figures and three tables it is unclear to us  how to proceed. Could you further clarify  your suggestions ?

In addition, the  standard of English has been checked and improved.

Please do not hesitate to inform me of any further changes which may be required.

Thank you for all your help.

Best regards,

Luz Herrero Manley

Reviewer 3 Report

Dear authors, the recommendation is that in the introduction, you specify precisely what novative/innovative elements this article presents and their place in the literature.  Regarding similarity coefficients, please kindly operate all elements of rewording/replacement based on the attached report.

1.      Please respect the journal's bibliographic reference style:

In the text, reference numbers should be placed in square brackets [ ], and placed before the punctuation; for example, [1], [1–3] or [1,3].  For embedded citations in the text with pagination, use both parentheses and brackets to indicate the reference number and page numbers; for example, [5] (p. 10).  or [6] (pp. 101–105).

2.      Lines 48-49.  There is a minor formatting problem.  Please correct it.

3.      I understand there are two research aims: For this reason, the main objective of this study was to verify the relevance and accuracy of diagnosing EKOA according to the existing criteria.

In addition, the secondary objective was to evaluate the clinical progression of healthy subjects at risk of OA, as well as patients with EKOA.  Please kindly insert in one paragraph the two purposes of the research.  Before the paragraph with the two aims, please insert this study's novel or innovative elements.

4.      It is recommended to introduce sample size calculation, statistical power or both. (Lu N, Han Y, Chen T, et al. Power analysis for cross-sectional and longitudinal study designs.  Shanghai Arch Psychiatry.  2013;25(4):259-262. doi:10.3969/j.issn.1002-0829.2013.04.009)

5.      In the discussion part, please put more emphasis on comparing your results with other similar studies.

6.      In terms of similarity, the article has 27%; not a big percentage problem, but a huge problem because 18% of the coefficient is from a single source/article.  

Moderate editing of the English language is needed.

Author Response

Dear Sir/Madam,

Thank you for your suggestions for improving the paper.

We have taken them into consideration and made changes according to your suggestions.

In the Introduction section we have clarified the innovative aspect  of this article and corrected the reference mistakes.  We have also explained both aims of the study in the same paragraph.

Regarding the simple size calculation it was not possible to calculate  this, as this paper is part of a wider project and the paper rose from the idea of analizing the EKOA classification criteria.

Regarding the similarity percentage problem , this could be due to the fact that  before we sent the paper to your journal we  had sent it to others and  one of them uploaded an online preprint.

In addition, the  standard of English has been checked and improved.

Please do not hesitate to inform me of any further changes which may be required.

Thank you for all your help.

Best regards,

Luz Herrero Manley

Round 2

Reviewer 3 Report

Dear authors, the similarity coefficient report I sent you specifies which article this problem comes from: https://www.ncbi.nlm.nih.gov/pmc/articles/PMC9689265/.  This is not a preprint of any kind, but an already-published article of yours, so there are similarities linked a self-plagiarism.  I am also attaching the article: Alabajos-Cea A, Herrero-Manley L, Suso-Martí L, et al.  Screening Clinical Changes for the Diagnosis of Early Knee Osteoarthritis: A Cross-Sectional Observational Study.  Diagnostics (Basel).  2022;12(11):2631. Published 2022 Oct 30.  doi:10.3390/diagnostics12112631.

While the sample size calculation part can be overlooked because you have included it in the article, Effect sizes (d) were calculated according to Cohen’s method, in which the magnitude of the effect was classified as small (0.20–0.49), moderate (0.50–0.79) or large (0.8), the similarity part requires the changes suggested initially.  The article is valuable; the rest of the suggested changes have been made, and we need to eliminate any problems that may arise before it is published.  So please kindly make the changes as initially suggested.

Author Response

Dear Sir/ Madam,

Thank you again for your suggestions for improving our paper.

We have taken them into consideration and made changes according to your suggestions.

Regarding the similarity coefficient report with an already-published article of ours, we have tried to make some changes in order to correct this.

 I would like to point out that both articles come from the same study although they focused on different aspects. Consequently, there is an overlap in the Materials and Methods section which I have tried to overcome.

Please do not hesitate to inform me of any further changes which may be required.

Thank you for all your help.

Best regards,

Luz Herrero Manley

Round 3

Reviewer 3 Report

Dear authors,

Congratulations on the excellent work you have undertaken in producing this scientific material.

Minor editing of English language required.

Author Response

Dear Editor,

Thank you for your suggestions for improving our paper and for the opportunity of publishing in your journal.

We have taken them into consideration and made changes according to your suggestions.

Reference style has been corrected to a ACS style as suggested in your instructions for authors.

All the other mistakes have also been corrected

Please do not hesitate to inform me of any further changes which may be required.

Thank you for all your help.

Best regards,

Luz Herrero Manley